# Differentially Expressed Somatostatin (SST) and Its Receptors (SST1-5) in Sporadic Colorectal Cancer and Normal Colorectal Mucosa

**DOI:** 10.3390/cancers16213584

**Published:** 2024-10-24

**Authors:** Agnieszka Geltz, Agnieszka Seraszek-Jaros, Małgorzata Andrzejewska, Paulina Pietras, Marta Leśniczak-Staszak, Witold Szaflarski, Jacek Szmeja, Aldona Kasprzak

**Affiliations:** 1Department of Histology and Embryology, Poznan University of Medical Sciences, Swiecicki Street 6, 60-781 Poznan, Polandmandrzej@ump.edu.pl (M.A.); paulina.pietras@student.ump.edu.pl (P.P.); 85124@student.ump.edu.pl (M.L.-S.); witold@ump.edu.pl (W.S.); 2Doctoral School, Poznan University of Medical Sciences, Bukowska Street 70, 60-812 Poznan, Poland; 3Department of Bioinformatics and Computational Biology, Poznan University of Medical Sciences, Bukowska Street 70, 60-812 Poznan, Poland; seraszek@ump.edu.pl; 4Department of General and Endocrine Surgery and Gastroenterological Oncology, Poznan University of Medical Sciences, Przybyszewski Street 49, 60-355 Poznan, Poland; jszmeja@ump.edu.pl

**Keywords:** non-neuroendocrine colorectal cancer, control colorectal mucosa, somatostatin, SST1-5, immunohistochemistry, RT-qPCR, pathogenesis, diagnosis and prognosis

## Abstract

Somatostatin, a growth hormone-release-inhibiting peptide (SST/SRIF), is a widely distributed multifunctional cyclic peptide that acts by activating G protein-coupled receptors (SSTRs, SST1-5). The diagnostic and prognostic role of these peptides in sporadic CRC remains unclear. The current study showed significant differences in the expression of SST and all SSTRs in the tissue material of sporadic CRC, control colorectal mucosa, and in lymph node metastases from the same patients. A significant dependence of the immunoexpression of SST1-5 on the cellular localization pattern was proved. Differences and/or significant correlations of SRIF system components with clinicopathological data (age, histological subtype of CRC, TNM parameters, Ki-67 antigen, and laboratory tests) were found. Interestingly, only control tissue showed differences in SST1-5 expression depending on the colon segment. The coexpression of all SST1-5 and overexpression of not only SST2 and SST5 in CRC may have applications for future therapy based on the SRIF system in sporadic CRC.

## 1. Introduction

Colorectal cancer (CRC) is one of the most common human malignancies worldwide, ranking third in terms of incidence and second in terms of mortality [1]. Genetic/epigenetic and environmental factors contribute to the onset and development of CRC [2,3]. Among the attractive hypotheses for the initiation and growth of CRC is the cancer stem cell (CSC) concept hypothesis, which is based on the presence of a small subpopulation of cells with embryonic stem cell characteristics [4]. There is evidence of close cooperation between CSCs and neuroendocrine cells (NCs), which are located adjacent to colonic SCs in the crypt SC niche [5,6]. Somatostatin, a growth hormone (somatotropin)-release-inhibitory factor (SST/SRIF) plays an important role in maintaining CSCs in a quiescent state [5,7].

SST acts by activating of five membrane receptors (SSTRs, SST1-5), belonging to the G protein-coupled receptor (GPCR) family [8,9]. In physiology, SST plays the role of a typical neurotransmitter/neuromodulator in the central nervous system. In peripheral tissues and especially in the gastrointestinal tract (GIT), it plays the role of a pan-inhibitory peptide in endocrine and exocrine secretion processes. It also affects motility, blood flow, and intestinal absorption, as well as exhibiting anti-inflammatory activities [10]. The SRIF system is also characterized by a strong antitumor activity, increasing cell apoptosis and inhibiting angiogenesis in most cancer tissues [7,9,11,12,13,14]. Thus, SSTR subtypes are crucial in the diagnosis and prognosis of many cancers (including CRC) and may be used in the therapy of these cancers [8,9,10].

Among GIT tumors, direct and indirect antitumor effects of SSTs have been best documented in gastroenteropancreatic neuroendocrine tumors (GEP-NETs) [15,16,17]. However, further studies are needed to determine the role of the SRIF system and its mechanism of action in the pathogenesis of non-NETs in the GIT, including sporadic CRC [18]. Remaining questions include, e.g., the role of the SRIF system in the histogenesis of colorectal adenocarcinoma cells [15,19,20,21,22], the origin of signet ring cells in mucinous adenocarcinoma [23], and SC overpopulation [5]. Similarly, the mechanisms of immune system control via the SRIF system in CRC are also poorly understood [14,24]. Furthermore, the diagnostic–prognostic role of SST/SSTRs tissue expression in sporadic CRC is still unclear. Recently, the role of the epigenetic mechanisms in modifying the expression of the SRIF system has been widely discussed [25,26]. Methylated SST is one of the biomarkers for early detection of CRC [25]. Studies on tissue expression of SST in CRC are few and inconsistent. Moreover, only some of them include the simultaneous examination of cancer tissue and histologically normal colonic mucosa from the same patient. SST-immunoreactive structures include enteroendocrine cells (EECs) (including D cells), carcinoma cells, adjacent crypt cells [12,20,21,27,28], as well as neurons and nerves in the colonic enteric nervous system (ENS) [29].

The differential tissue expression of SSTRs studied in NETs, mixed adenoneuroendocrine carcinomas and pure colorectal adenocarcinoma affects both morphologically normal colonic mucosal cells and primary and/or metastatic tumor tissues [5,15,30,31,32,33,34,35].

In recent years, the prognostic significance of tissue marker expression in normal colorectal mucosa adjacent to the tumor (NAT) has also been emphasized [36,37]. NAT is commonly used as a tissue control in cancer research. The Cancer Genome Atlas (TCGA) dataset guidelines specify that NAT samples must be collected more than 2 cm from the tumor margin and/or the tissue must be histopathologically confirmed to be free of cancer cells [36]. This is important because some transcriptomic studies suggest that NAT may represent a unique intermediate state between healthy and tumor tissue type. More than a dozen genes have been shown to be specifically activated in NAT, and only the mechanisms that alter the expression of these genes remain to be verified [38]. Using current NAT criteria, a prognostic role for high levels of the cancer stemness-related gene and protein, chromobox 8 (CBX8) in NAT (worse DFS and OS) has been demonstrated [36]. Similarly, miR-509-3p expression in NAT tissue was an independent prognostic factor in CRC [37].

This study aimed to determine the role of tissue expression of SST and SST1-5 in the pathogenesis, diagnosis, and prognosis of sporadic CRC. Understanding the role of the SRIF system in CRC may help to identify new diagnostic and therapeutic targets with new forms of SST analogs (SSAs) in this cancer.

## 2. Materials and Methods

### 2.1. Patients and Tissue Samples

This study included 34 CRC patients (28 men, 6 women) who underwent surgery and were not treated with other methods (radio- or chemotherapy) from the Department of General and Endocrine Surgery and Gastroenterological Oncology, Poznan University of Medical Sciences, Poland. Patients’ clinical data were obtained before surgery. Most of the patients in this CRC group had been subject to other analyses by our team [39,40]. The available clinical data included histopathological diagnosis; tumor size, i.e., flat (<3 cm in diameter) or protruding (≥3 cm); tumor localization, i.e., proximal colon (right colon), distal colon (left colon), and rectum, and/or entire colon vs. rectum; grade and stage according to Dukes’ classification, Astler and Coller’s modified Dukes’ classification, and the TNM system [41,42]; age; sex of the patient; and basic laboratory tests (e.g., blood count, leukocytes, and glucose levels). Patient survival time reflected the time between the surgery date (1 October 2010) and 1 October 2015. Patient characteristics are shown in Table 1.

Thirty-four paired samples of CRC and non-tumor tissues were obtained during surgical treatment. For molecular analysis, the tissue samples (~0.5 cm^3^) were collected and stored in RNAlater^®^ (Applied Biosystems (Carlsbad, CA, USA)) at −80 °C until used for mRNA expression analysis and/or fixed in 10% buffered formalin and embedded in paraffin for histological assessment. Control tissue for RT-qPCR analysis included colorectal mucosa and, depending on the depth of tumor invasion, submucosal layers at a maximum distance of 10–15 cm from the diseased area from the same patient. Control tissue for immunohistochemical (IHC) analysis in our patients and tissue microarray (TMA) specimens (Section 2.2) included morphologically normal colorectal tissue adjacent to the tumor (NAT) at a minimum distance of 2 cm from the tumor margin in the same patients. All control tissues met the current NAT criteria [36]. Formalin-fixed, paraffin-embedded colorectal tissue from patients without CRC; the normal thyroid gland of mature rats; and neuroendocrine lung carcinoids were used for qualitative (not quantitative) comparative IHC analysis (positive control).

### 2.2. The Tissue Microarray (TMA)

In addition, commercially available TMA slides obtained as unstained sections (CO992a and CO992b, BioCat GmbH, Heidelberg, Germany) containing core samples of CRC (*n* = 33) with matched lymph node metastasis (LNM) and NAT were used for the IHC study. This part of the study did not require the local bioethics committee’s approval. The characteristics of the CRC patients included in the TMA are shown in Table 2.

### 2.3. Real-Time Quantitative PCR (RT-qPCR)

CRC fragments and control colorectal mucosa from only 25 patients were selected for RT-qPCR analysis because not all 34 patient samples yielded sufficient RNA quality in all primary tumor sites and/or control tissues. The steps of the technique have been described previously [43]. Total RNA was isolated according to the protocol for the Universal RNA/miRNA Purification Kit (EURx, Gdańsk, Poland) and quantified using NanoDrop (Thermo Fisher Scientific, Waltham, MA, USA). LunaScript RTTM SuperMix (New England Biolabs (Ipswich, MA, USA), #10105340) generated cDNA according to the manufacturer’s guidelines. RT-qPCR was performed using Luna^®^ Universal qPCR Master Mix (New England Biolabs, #10103269) in the CFX96 Real-time PCR System (Biorad (Hercules, CA, USA)) according to the manufacturer’s instructions. The PCR protocol was as follows: (1) initial denaturation, 95 °C, 60 s; (2) denaturation, 95 °C, 15 s; (3) annealing and extension 60 °C, 30 s. The number of cycles was 40. We chose glyceraldehyde-3-phosphate dehydrogenase (*GAPDH*) and hypoxanthine phosphoribosyltransferase 1 (*HPRT1*) as reference genes. Evaluation of changes in SST and SST1-5 mRNA expression involved comparing mRNA copy numbers for the SST and SST1-5 per microgram of RNA between the tumor and control samples from the same patient. Absolute quantification determines the exact copy number of a target gene by relating the *C*_t_ value to a standard dilution curve of total mRNA. Before absolute quantification, the *C*_t_ values were normalized compared to the average of *C*_t_’s obtained for two housekeeping genes (*GAPDH* and *HPRT1*). The primers for RT-qPCR assay are listed in Appendix A.

### 2.4. Immunohistochemistry (IHC)

Paraffin tissue sections from all the patients, 5 μm thick, deposited onto SuperFrost/Plus microscope slides, as well as TMA sections after deparaffinization and rehydration through a series of decreasing ethanol concentrations, washed in phosphate-buffered saline (PBS), were subjected to routine steps of the IHC protocol [44].

Somatostatin mouse monoclonal antibody (Ab) (clone ICDCLS) (eBioscience™, Invitrogen™, Affymetrix eBioscience 14-9751-80 (Fisher Scientific)) was used at a dilution of 1:50. Rabbit polyclonal antibodies specific for SST1 (orb229340, at 1:500 dilution), SST2 (orb221903, at 1:500 dilution), SST3 (orb11423, at 1:50 dilution) (all Abs from Biorbyt Ltd., Cambridge, CB5 8LA, UK), SST4 (DF2779, at 1:200 dilution) (Affinity Biosciences (Middlesex, UK), www.affbiotech.com), SST5 (orb11424, at 1:200 dilution) (Biorbyt Ltd.), and Ki-67 antigen (clone MIB-1) (Dako Denmark A/S, Glostrup, Denmark, ready to use) were used. Sections were incubated with these primary Abs overnight, at 40 °C, followed by incubation with dextran conjugated to horseradish peroxidase (HRP) and secondary biotinylated conjugated anti-rabbit and anti-mouse IgG (Dako REALTMEnVisionTM Detection System peroxidase/DAB+, Rabbit/Mouse, Dako). In at least three consecutive sections, positive reaction was manifested as a dark brown or black precipitate in the cell nucleus (Ki-67) and cell membrane/cytoplasm/nucleus (SRIF system components). The slides were counterstained using Mayer’s hematoxylin (#S330930-2, Dako).

Each test was accompanied by an internal negative control, in which specific Abs were replaced with a normal serum of a corresponding species in 0.05 M Tris-HCl, pH~7.6, supplemented with 0.1% bovine serum albumin (BSA) and 15 mM sodium azide (Sigma-Aldrich, St. Louis, MO, USA).

Histological slides with IHC expression in the tumor and control colorectal mucosa were examined using the Olympus BH-2 optical microscope coupled to a digital camera. Color microscopic images were captured and archived using a 40× objective (at least 10 fields in each slide with an IHC positive reaction) using LUCIA Image 5.0 computer software.

#### 2.4.1. Semiquantitative Evaluation of IHC Expression

Expression of Ki-67 antigen in clearly labelled cell nuclei was calculated, as the mean percentage of immunopositive cells in 10 light microscope fields. Expression was evaluated using the modified semi-quantitative scale as previously described, in which a score of 1 corresponded to up to 10% of positive cells, and scores of 2, 3, and 4 corresponded to 11–25%, 26–50%, and ≥51% of positive cells, respectively [43].

#### 2.4.2. Morphometric Evaluation of SST and SST1-5 Expression

The images with positive IHC reaction, 2560 × 1920 pixels in size, recorded using LUCIA Image 5.0 software, were subjected to morphometric analysis using the quantitative morphometric Filter HSV software, originally developed at the Department of Bioinformatics and Computational Biology, Poznan University of Medical Sciences, according to the following formula: A% = area of IHC reaction (pixels)/tissue area (pixels) × 100% [40].

### 2.5. Statistical Analysis

Statistical analyses were performed using Dell Statistica, version 13 (Dell Inc., 2016; Palo Alto, CA, USA). Quantitative data are presented as mean, median, minimum, and maximum values; lower and upper quartiles; and standard deviation. The results were first checked for normality using the Shapiro–Wilk test. Since the test confirmed a lack of normality, non-parametric tests were used. The Mann–Whitney U test was used to compare two groups, and the Kruskal–Wallis test with multiple comparisons as a post-hoc analysis was used to compare more than two groups. Comparative analysis of SST1-5 expression was performed using the Friedman test with Bonferroni post-hoc correction, separately, in the CRC, LMN, and control groups. Additionally, Spearman’s rank correlation coefficients were calculated to assess the relationships between selected quantitative variables. The percentages of IHC positivity of SST and SSTRs were evaluated, using the difference test between two proportions. The dependence of SST1-5 detection on the cellular localization of the marker in each studied group was assessed using the Chi-square test. Two subgroups of tumor samples were also determined: below the mean expression and above the mean expression of IHC reactions for SST and SST1-5. The survival time of patients in both determined groups was analyzed by Kaplan–Meier and log-rank test. These analyses were performed using MedCalc^®^ Statistical Software version 20.123 (MedCalc Software Ltd., Ostend, Belgium; https://www.medcalc.org; 2022). Results were considered significant at a *p*-value of less than 0.05.

## 3. Results

### 3.1. Quantitative Analysis of SST and SST1-5 mRNA Expression in CRC and Control Mucosa Samples

The SST mRNA was significantly lower in CRC when compared to control samples (*p* < 0.0001). In contrast, SST2, SST3, and SST5 transcripts showed higher levels in CRCs compared to controls. SST4 mRNA levels were also borderline higher in tumor than in control (*p* = 0.059) (Table 3).

In both CRC and controls, the SST4 transcript dominated the expression of all other transcripts (*p* < 0.001) (Appendix A).

### 3.2. Reciprocal Expression of SST and SST1-5 mRNA

In both CRC and control tissues, we observed high correlations between the reciprocal expression of the following SSTR transcripts: (1) SST2 with SST3 and SST4; (2) SST3 with SST2, SST4, and SST5; (3) SST4 with SST2 and SST3. Additionally, a highly significant Spearman’s correlation was found between the reciprocal expression of SST mRNA and SST1 mRNA in the control group (R = 0.69, *p* < 0.05) (Table 4).

### 3.3. Immunoexpression Frequency and Tissue Localization of SST and SST1-5

#### 3.3.1. Somatostatin (SST)

There were no significant differences (*p* > 0.05) between the incidence of SST expression in CRC tissue samples and histologically normal colorectal tissue (C) (88%) (Table 5).

The malignant cells with a cytoplasmic product of the SST IHC reaction were demonstrated to be focally localized (Figure 1A), with a granular reaction pattern in the apical part of the cancer cells (Figure 1B), or distributed throughout the cell cytoplasm (Figure 1C). In control colorectal tissue, individual SST-immunoreactive cells (EECs) were observed in addition to other immunopositive epithelial cells (Figure 1D–F).

IHC analysis of control colorectal crypts from the patients without CRC showed isolated SST-positive cells (Figure 1G). C cells of the normal thyroid gland (Figure 1H) and the disseminated endocrine cells of a typical lung carcinoid (Figure 1I) were also SST-positive.

#### 3.3.2. Somatostatin Receptors (SST1-5)

The positive rate of SST1 expression was 100% of cases in all tested groups (CRC, LNM, and C) (Table 5). In CRC, the positive rate of SST1 was notably higher compared to SST3, and SST4 (Figure 2A). No significant differences in the detection rates of SST1-5 expression were found in the LNM group (Figure 2B). In controls, it was higher compared to SST2 and SST3 (Figure 2C).

In CRC, the predominant IHC reaction was present in tumor cells, although SST1 positivity was also observed in tumor stromal cells. In control mucosa, SST1 expression was observed in epithelial cells and cells of the lamina propria (Figure 3).

The percentage of different forms of cellular localization (patterns of IHC reaction) of SST1-5 in patients from all groups is shown in Figure 4.

Our study revealed a dependence of SST1-5 detection rate on the cellular localization of the marker in each studied group (Table 6).

The difference test between two proportions was used to demonstrate quantitative differences between the percentage of the IHC reaction pattern of a given receptor (Appendix A). Regarding the detection of IHC expression pattern of SST1 in CRC, a cytoplasmic pattern (pattern 1) was significantly higher compared to SST2 and lower compared to SST4. In LNM, the mixed IHC pattern (pattern 2) of SST1 expression was predominant and significantly higher compared to SST3, SST4, and SST5. In control tissues, pattern 2 of SST1 expression was predominant, but was only higher compared to SST4 (Appendix A).

The frequency of SST2 expression in CRCs and LNM was also high and differed significantly in both groups compared to controls (Table 5). In addition, the frequency of SST2 expression was significantly higher than SST3 in CRC and higher than SST5 in controls (Figure 2). It is noteworthy that for SST2, the highest percentage of tissues with pattern 3 of IHC reaction (with predominance of membranous over cytoplasmic IHC reaction or clear membranous pattern) was observed in CRC and control groups among the other SSTRs (Appendix A).

Regarding SST3 expression, the percentage of lowest detection rate of this receptor was shown in CRC (Table 5), significantly lower compared to the presence of SST1, SST2, and SST5 (Figure 2). Only tissues with LNM showed a 97% prevalence of SST3 expression, but no significant differences between receptor detection in this group of patients. In controls, the frequency of SST3 detection was significantly lower compared to SST5 (Figure 2). SST3 expression was present in tumor cells, morphologically normal intestinal epithelial cells, as well as in tumor stromal cells and lamina propria of control mucosa (Figure 3). In CRC and LNM, the cytoplasmic IHC pattern (pattern 1) was predominant, whereas in control mucosa, pattern 2 prevailed (Table 6). In CRC, the detectability of pattern 1 was significantly higher compared to SST2. In LNM, the percentage of SST3 with pattern 1 was higher compared to SST1 and SST2 (Appendix A).

The incidence of SST4 in CRC, LNM, and controls was similar (Table 5). In CRC, a significant difference only observed between the frequency of SST4 and SST1 (Figure 2). The cytoplasmic reaction pattern (pattern 1) for SST4 was predominant in all groups studied (Table 6). SST4 expression was significantly higher in CRC with pattern 1 compared to SST1, SST2, and SST5. Similarly in control tissue, SST4 expression was significantly higher in pattern 1 compared to SST1, SST2, SST3, and SST5 (Appendix A). SST4 immunoexpression was demonstrated by both tumor and histologically normal epithelial cells. The LNM group showed SST4 expression in tumor cells and numerous lymphocytes (Figure 3).

In turn, heterogeneous expression of SST5 was observed mainly in tumor cells (Figure 3). Its positive rate in CRC and LNM was 100%, with a predominantly cytoplasmic pattern of IHC reaction (Appendix A). In control mucosa, the presence of SST5 expression was demonstrated in 98% of tissues (Table 5), whereas a mixed IHC pattern predominated compared to SST2 and SST4 (Table 6). Significant differences were observed between SST5 and SST3 in CRC and between SST5 and SST2 and SST3 in control mucosa (Figure 2). Considering the detection rate of pattern 1 of SST5 in CRC, it was significantly higher than SST2, and lower than SST4 (Appendix A).

### 3.4. Quantitative Analysis of SST and SST1-5 Immunoexpression

SST immunoexpression was significantly lower in the tumors than in the controls. There were also statistically significant differences in the expression of SSTRs (except for SST3) between the different study groups (Table 7). With regard to SST1, significantly higher expression was observed in LNM and CRC vs. controls (*p* < 0.0001, *p* = 0.0007, respectively). SST2 expression was also highest in LNM, followed by CRC, and both were significantly higher than in control mucosa (*p* < 0.0001 in both cases). In contrast, SST3 expression was quantitatively similar in CRC, LNM, and control mucosa. Regarding SST4, there was significantly higher expression of this receptor in LNM compared to CRC (*p* = 0.003) and with expression in control mucosa (*p* = 0.0005). For SST5, significant differences were found only between expression in CRC and control samples (*p* < 0.0001) (Table 7).

#### 3.4.1. Comparison of the Mutual Tissue Immunoexpression of SST1-5 within Groups

In CRC, LNM, and controls, significant differences were observed in the expression of SST1-5 (Appendix A). The CRC showed significantly higher IHC expression of SST1 than SST3 and SST4. In addition, SST2 expression was higher than SST3 and SST4, and SST5 expression was greater than SST3 and SST4. In LNM, SST1 expression was also observed to be higher than SST3 and SST4. The expression of SST2 and SST5 was significantly higher compared to SST3. In controls, significantly higher expression of SST1 was observed compared to SST2, SST3, and SST4. Similarly, higher expression of SST5 over SST2, SST3, and SST4 was noted (Figure 5).

#### 3.4.2. Correlation between Mutual Immunoexpression of SST and SST1-5 in Each Group

In CRC samples, SST expression did not correlate with the expression of any SSTR. However, SST expression evaluated in control mucosa samples showed positive correlation with SST5 expression. In each study group, many high correlations were observed between the reciprocal expressions of SST1-5 (Table 8).

#### 3.4.3. Comparison of SST1-5 Expression in Relation to Pattern of IHC Reaction

The CRC group showed significantly higher expression of SST3 with a mixed reaction pattern (pattern 2) compared to the cytoplasmic IHC reaction pattern (pattern 1) of this receptor. In contrast, the LNM group showed significantly higher expression of SST2 with pattern 2 vs. pattern 1. The most significant differences in tissue expression regarding IHC reaction patterns were observed in the control mucosa. This was true for all SSTRs, except for SST4. For SST1, SST3, and SST5, there was a predominance of pattern 2 expression over pattern 1. For SST2, there were differences between expression with pattern 1 and 2 vs. pattern 3 (*p* = 0.026, *p* = 0.032, respectively) (Table 9).

#### 3.4.4. Correlation Between mRNA and Peptide Expression of SST and SST1-5

No significant correlations were found between mRNA and peptide expression for SST in any of the groups examined. In CRC, a positive Spearman’s correlation was found between mRNA and peptide expression of SST1 (R = 0.52; *p* < 0.05). SST4 mRNA also correlated with SST1 immunoexpression (R = 0.51; *p* < 0.05). In both examined groups, we observed negative correlation between SST2 mRNA and SST3 immunoexpression. Additionally, SST3 mRNA correlated significantly with SST1 immunoexpression in the control group (Table 10).

### 3.5. SST and SST1-5 Expression and Pathological Data

#### 3.5.1. Macroscopic Type of the Tumor (Flat vs. Protruding)

No significant differences in SST and SST1-5 (mRNA and peptide) expression in CRC samples were found between macroscopic tumor types.

#### 3.5.2. Tumor Localization (Colon vs. Rectum and Proximal, Distal Colon and Rectum)

No significant differences in SST and SST1-5 mRNA expression were observed in CRC samples in the colon vs. rectum localization of the tumor, nor in the different anatomical locations of the tumor (proximal, distal, and rectum). In control tissues, there were no differences in the expression of SST and SST1-5 mRNA in the colon vs. rectum localization, as well as in the right colon, left colon, and rectum localization (*p* > 0.05).

No significant differences in SST and SST1-5 immunoexpression were shown in CRC samples in the colon vs. rectum localization of the tumor, nor in the different anatomical locations of the tumor (proximal, distal, and rectum) (Table 11). In control tissues, SST3 immunoexpression was significantly higher in histologically normal rectums (14.74 ± 9.03%) compared to normal colons (8.88 ± 9.01%) (*p* = 0.033). The control tissues showed significantly higher expression of SST2 and SST4 in rectums compared to distal colons (*p* = 0.004, *p* = 0.01, respectively), and higher expression of SST3 in rectums compared to both proximal and distal colons (*p* = 0.02, *p* = 0.002, respectively) (a multiple comparisons test) (Table 11).

#### 3.5.3. Histological Stage of the Tumor

No significant differences in SST and SST1-5 (mRNA and peptide) expression were found between tumors of different Dukes’ stages, Astler–Coller classification stages B2 vs. C2, or in different clinical TNM clinical stages (II vs. III as most common in CRC patients). When comparing the analyzed SST and SSTRs based on tumor size (T parameter) at diagnosis, significant differences were found only for SST2 immunoexpression (*p* = 0.017) (Kruskal–Wallis test). Significantly higher SST2 expression was observed in patients with T4 compared to T3 and T2 groups (*p* = 0.036, *p* = 0.030, respectively) (a multiple comparisons test). There were no significant differences in SST and SST1-5 expression between patients without (LNM0) and those with lymph node metastasis (LMN1-2). Regarding distant metastasis, Mann–Whitney’s U test could only be applied in the case of SST, SST3, and SST5 immunoexpression. Significantly higher immunoexpression of SST3 was observed in CRC with distant hepatic metastasis (40.1 ± 17.7%) in comparison with patients without metastasis (14.7 ± 18.0%) (*p* = 0.012).

#### 3.5.4. Histological Grade of the Tumor

No significant differences of SST and SST1-5 (mRNA and peptide) expression were found between tumors of different malignancy grades (G1, G2, and G3) (*p* > 0.05).

#### 3.5.5. Nonmucinous vs. Mucinous CRC Subtype

No significant differences in SST (mRNA and peptide) expression were observed between mucinous and nonmucinous CRC subtypes (Table 12 and Table 13). Considering the SST1-5 mRNA expression, significantly higher SST3 mRNA expression was detected in mucinous as compared to nonmucinous CRC subtypes (Table 12).

Considering SST1-5 immunoexpression, significantly lower SST1, SST2, SST3, and SST4 expression was detected in the mucinous CRC subtype, compared to nonmucinous tumors (Table 13).

#### 3.5.6. Ki-67 Expression and Its Correlation with SST and SST1-5 Expression

A significantly higher expression of Ki-67 was detected in the CRC samples (2.55 ± 1.42, median 3.0) compared to the control samples (1.31 ± 1.00, median 1.00) (*p* = 0.0004). No significant correlation was observed between SST (mRNA and peptide) and Ki-67 antigen expression. Among SST1-5, only a low negative correlation was found between SST3 immunoexpression and Ki-67 (R = −0.38, *p* = 0.039).

### 3.6. SST and SST1-5 Expression and Selected Clinical Data

We observed a moderate negative correlation between SST mRNA and age in the CRC and control group (R = −0.42 in both cases, *p* < 0.05). No significant correlation was observed between SST1-5 mRNA expression and age, and basic laboratory tests in CRC patients. In the control group, a moderate positive correlation between SST1 mRNA and leukocyte count was noted (R = 0.44, *p* < 0.05) (Table 14).

Immunoexpression of SST in CRC and control mucosa samples was not significantly correlated with age and basic laboratory tests. In both groups (CRC and C) a low negative correlation between immunoexpression of SST3 and SST4 and age was observed, while in control mucosa, SST2 expression was negatively correlated with age (R = −0.45, *p* < 0.05, moderate correlation) (Appendix A). With regard to sex differences, the Mann–Whitney U test could only be applied in the case of SST and SST1-5 immunoexpression (not mRNA expression). However, there were no differences between the expression of SST and five SSTRs and sex.

The mean survival time of CRC patients was 45.4 ± 16.5 months (Table 1). Kaplan–Meier analysis showed that neither immunoexpression of SST nor any of the SSTRs in tumor samples was significantly associated with the survival probability of CRC patients (Figure 6). For the prognostic role of SST and SST1-5 mRNA, statistical analysis was not possible due to insufficient group size. There was also no correlation between survival time and immunoexpression of the tested markers in control tissues.

## 4. Discussion

### 4.1. Main Findings on Somatostatin

The high frequency of SST immunoexpression in the CRCs examined in our study, as well as in control samples (88%), confirms the results of two other studies [19,20]. However, there are also studies showing a very low percentage of SST-immunopositive CRCs (7%) [21] or no expression of this neuropeptide in primary mucinous CRC [23]. Unlike other authors [12,21], we observed a more heterogeneous SST immunoexpression, ranging from single SST-positive cell EECs to a strong IHC reaction present in numerous cells of altered adenocarcinoma structures, or within the stroma of the tumor.

Using quantitative methods we demonstrated a significantly lower SST expression (mRNA and peptide) in CRC compared to controls, confirming the observations of other authors [12]. However, there are also studies demonstrating the presence of SST mRNA only in control samples and not in CRCs [5]. We were unable to demonstrate significant correlations between SST (mRNA and peptide) expression in the tissue material we examined. Interestingly, SST transcript (but not peptide) expression was negatively correlated with patient age in both CRC and controls. Thus, we cannot confirm the part of the study by Leiszter et al. in which they found no age-dependent changes in SST mRNA production in the natural ageing process of the organism [12]. However, it seems that the decrease in SST production in histologically normal colorectal tissue and CRC patients with age may indicate a putative role for SST in the pathogenesis of CRC and may be a risk factor for this cancer.

In the current study, there were no differences in SST expression (mRNA and peptide) regarding the clinicopathological data (including survival probability), Ki-67 antigen expression, and/or basic laboratory tests. A high positive correlation was observed between SST mRNA and SST1 mRNA expression, but only in control tissues (R = 0.69, *p* < 0.05). Other studies have documented the highest levels of SST peptide (mRNA not studied) in well-differentiated CRC tissues compared to poorly differentiated tumors. Low SST expression (along with high glucagon expression and the presence of gastrin) would be expected to indicate a poor prognosis [45].

### 4.2. Main Findings on Somatostatin Receptors

SST receptors exhibit a broad spectrum of biological activities, including effects on secretion of hormones, such as growth hormone, insulin, glucagon, as well as immune response [8,9,10]. The occurrence of a high density of SSTRs has prognostic significance and may be used in cancer therapy [8,9,10,17]. To prolong the biological activity of the SST peptide in circulation (half-life of about 3 min), its analogs are used, including octreotide (OCT) (90–120 min). Other SAAs are applied in radiolabeled SST analog/peptide receptor radionuclide therapy (PRRT) [24]. Moreover, while native SST binds to all SSTRs, OCT binds with high affinity to SST2 and SST5 [16]. SST analogs in vivo can regulate tumor growth through direct and indirect actions, inhibiting the secretion of GH, growth factors, and angiogenesis. Moreover, when SSTRs are coexpressed, they can interact to form homo- or heterodimers, including with other GPCRs, e.g., dopamine type 2 receptor and μ-opioid receptor 1, altering their original pharmacological and functional properties. The demonstration of coexpression of all SST1-5 may have applications for therapy based on the SRIF system [10,11,16,17].

#### 4.2.1. SST1 and SST5

In the current study, a fairly high significant correlation was obtained between SST1 mRNA and peptide expression in the CRC (R = 0.52, *p* < 0.05). Interestingly, SST1 mRNA correlated positively with SST mRNA expression in control tissues. In addition, only controls showed a significant positive correlation between SST1 transcript (but not peptide) expression and leukocyte levels in patients’ blood. Studies by other authors using RT-PCR and in situ hybridization (ISH) showed a highly variable expression of SST1 mRNA in CRC, ranging from sporadic [31], present in all control samples and in three out of five tumor samples [5], to frequently detected in normal and pathological colons [32]. Similar to our study, no correlation of SST1 mRNA expression with clinical data was observed [31]. The use of molecular biology techniques confirms the rather heterogeneous expression of SST1 mRNA in CRC, liver metastasis, and control mucosa [30]. Using ISH, it has been shown that SST1 mRNA expression in colon tissue is not restricted to NECs or clusters of immune cells in the lamina propria and stroma near the tumor, but occurs in morphologically normal mucosa as well as in tumor tissue [31,33]. We cannot comment on these observations from ISH studies.

The present study showed a significant overexpression of SST5 mRNA in CRC compared to control mucosa. However, it was significantly lower than SST4 mRNA expression in both study groups. In addition, positive correlations were observed between the reciprocal expression of SST5 with other SSTRs in both study groups. However, as for the SST1 transcript, no significant correlations of SST5 mRNA with clinicopathological data were noted. Other authors detected the SST5 transcript predominantly in early-stage tumors (tubular adenomas, Dukes’ A and B) compared to late-stage tumors (Dukes’ C and D tumors) [31]. Other studies, in agreement with our observations, did not show a correlation between SST5 mRNA expression and Dukes’ stage, thereby failing to confirm the loss of expression of this transcript (as well as SST2 mRNA) in later stages of CRC development. SST5 expression has been shown to be dominant among other receptors in both CRC and controls [32]. ISH studies have shown a high prevalence of SST5 mRNA in CRC and normal mucosa, with the SST5 mRNA hybridization signal being 2 to 4–5 times higher in tumor tissues than in control tissues [33].

Our IHC studies showed coexpression of all SSTRs throughout the tissue material. SST1 ranked first in terms of detection in all three groups, followed by SST5. The lowest percentage of positive tissue was for SST3 expression. In addition, a correlation between the expression of a given SSTR and cellular localization was observed in each of the groups studied. For SST1 and SST5 expression, the cytoplasmic pattern of IHC reaction (pattern 1) predominated in CRC. More diverse patterns of IHC reaction were observed in LNM and control groups. The presence of SST1 as the predominant SSTR type, and SST5 as the “second” in order (although in lower percentages of CRCs and controls than in our study), is also indicated by other authors [34].

In the quantitative analysis of SSTR immunoexpression, we confirmed high expression of SST1 and SST5 compared to the expression of other receptors in each of the groups studied. We also showed that SST1 expression was significantly higher in CRC and LNM, and SST5 expression was significantly higher in CRC compared to control mucosa. This analysis also showed more differences between the reciprocal expression of SSTRs in a given group than testing the detection of a given receptor alone. Interestingly, significantly lower expression of SST1 (but not SST5) was observed in the mucinous vs. nonmucinous subtype of CRC. However, we did not observe significant differences in SST1 and SST5 immunoexpression in relation to other clinicopathological data (including survival probability), Ki-67 antigen expression, and/or basic laboratory tests. In contrast to our study, other authors have shown more frequent SST1 expression in CRC with positive LNM than in those with negative LNM, and an increase in SST1 but a decrease in SST5 detection with increasing Duke’s stage. In addition, more SST5-immunopositive cases were observed in moderate-to-well-differentiated CRC than in poorly differentiated adenocarcinoma. However, no quantitative analysis of SSTR expression was performed in this study [34]. Instead, Evangelou et al. showed a negative correlation of SST5 expression with CRC invasion and liver metastasis, and it was the positive expression of SST5 (and SST2) that was a good predictor of survival in CRC [35]. We cannot confirm these study results.

#### 4.2.2. SST2

The study of human SST2 expression is important because it is the receptor subtype for which OCT, one of the first SSAs to be studied, has the highest affinity [9,10,18]. As shown for pancreatic NETs, SST2 has a higher affinity for SSAs than SST5 and SST3 [16]. It has been suggested that SST2 may be a potential predictive biomarker for response to treatment with the immune checkpoint inhibitors of many cancers, including CRC [46].

Our RT-qPCR studies showed a significant prevalence of SST2 transcript expression in the tumor compared to controls. However, the expression level of SST2 mRNA was one of the lowest among the other SSTR transcripts in both study groups, although statistical significance was only shown in comparison with SST4 mRNA. No significant changes in SST2 mRNA expression were observed depending on clinicopathological data. Other studies have indicated a loss of SST2 mRNA expression with tumor progression [30]. However, as with our study, correlations have not always been shown between SST2 mRNA expression and Duke’s stage [32]. Further studies by the latter authors using the ISH method confirmed only low expression of SST2/SST3/SST4, with a predominance of SST5 expression in CRC and control [33]. There are also studies showing similar quantitative expression of SST2 mRNA in CRC and non-cancerous intestinal tissue from the same patients [47,48]. No correlation with localization, grading, and stage of the tumor was observed. Loss of SST2 expression was found in patients with elevated preoperative CEA levels [48]. Further studies by these authors showed that patients with high SST2 mRNA expression had an unfavorable prognosis (an increase in cancer-related deaths) and significantly shorter disease-free survival [49].

The frequency of SST2 immunoexpression in the current study was significantly higher in CRC (98%) and LNM (100%) compared to controls (87%). Such differences in SST2 detection were not shown by Qiu et al. However, they observed more frequent expression in moderate-to-well-differentiated vs. poorly differentiated CRC [34]. In terms of IHC-positive rates of SST2, CRC was ranked third out of 20 different cancers, after carcinoid and thyroid cancer [46]. Noteworthy, in the current study, we observed the highest percentage of tissues with membranous or membrane-cytoplasmic localization of SST2 (pattern 3) compared to other SSTRs, and this was true in both CRC and control mucosa. Quantitative evaluation of the IHC reaction confirmed significantly higher SST2 immunoexpression in CRC and LNM vs. control. Additionally, we observed higher SST2 expression in patients with T4 vs. T2 and T3 parameters. Interestingly, SST2 expression was significantly lower in the mucinous subtype of CRC compared to the nonmucinous subtype. When these results are compared with the literature, there are some differences. Evangelou et al. showed negative correlations between SST2 (and SST5) expression levels and CRC invasion and liver metastasis. SST2 expression was significantly higher in lower-grade tumors and in tumors located in the rectum. Patients with SST2 (and SST5) expression had a better prognosis (had longer survival rates) [35]. We cannot confirm these observations. Other researchers even consider SST2 to be the most common SSTR, in both CRC and control samples [30,31,32,33,34,47,48]. Their study showed widespread expression of the human gene SST2 with about 90% detection in CRC and controls. Rather, homogeneous expression was found in tumor cells, control mucosal cells, and stromal cells in most of the samples analyzed. However, these authors did not observe a correlation of SST2 gene expression with clinical data [31].

#### 4.2.3. SST3 and SST4

It should be emphasized that the expression of SST4 mRNA significantly exceeded all transcripts of the tested SSTRs in CRC and control mucosa. Thus, our RT-qPCR studies do not confirm these findings of other authors, who showed rare (or complete absence of) heterogeneous [30,31], or low expression of SST3 and SST4 [33] in both CRCs and controls. In addition, positive correlations were observed between the reciprocal expression of SST3 mRNA and the other three transcripts, i.e., SST2, SST4, and SST5, in both study groups. In none of the tissue groups studied (CRC nor C) did we observe a correlation between the expression of SST3 and SST4 transcripts and the immunoexpression of the corresponding peptides. We demonstrated significantly higher expression of SST3 mRNA in the mucinous vs. nonmucinous CRC subtype. We did not find any other correlations with clinical data and the expression of these two transcripts, confirming some data from the literature [31].

Our study showed a high prevalence of immunoexpression of SST3 and SST4 in CRC, LNM, and control mucosa, higher than that reported by other authors [34]. However, we cannot confirm these authors’ observation of a decrease in detectable SST3 (and SST5) expression with increasing Dukes stage.

The quantitative IHC analysis performed in our study showed similar SST3 expression in all study groups. In contrast, SST4 expression was significantly higher in LNM compared to CRC and control. Thus, there were no differences in SST3 and SST4 immunoexpression between CRCs and controls. We showed a low negative correlation of SST3 and SST4 immunoexpression with age and in both CRC and control groups. The significance of these correlations is unclear and is not supported by the findings of other authors [34]. In addition, we showed lower expression of both of these SSTRs in the mucinous vs. nonmucinous subtype of CRC. Exclusively for SST3 immunoexpression in CRC, we showed a negative correlation with Ki-67 antigen expression. This may confirm, at least in part, the findings of other authors of a higher proliferative index in tissues with negative SST3 (and SST2) expression than in those with positive expression [34].

An interesting issue that needs to be addressed is the diagnostic and prognostic role of tumor localization in different segments of the colon. Studies suggest that right-sided colon cancer has a worse prognosis than left-sided CRC [50,51]. In addition, differences in the expression of various histochemical, IHC, or molecular markers are also shown in NAT located in different segments of the large intestine [36,40,52]. Our previous histochemical studies showed a higher expression of sulfomucins in the distal colon compared to the proximal segments [40]. In contrast, we found no such differences in the expression of IGF-1, MUC-1, or MUC-2 (mRNA, protein) [39,43]. Recent studies confirm differences in the expression of myeloperoxidase (MPO) protein and autophagy marker, microtubule-associated protein light chain 3 (MAPLC3), depending on the section of normal colorectal mucosa. Tissue expression of MAPLC3, especially in the right colon, correlates with the presence of adenomas [52]. In this study, we showed much more significant differences in SST1-5 expression in the NAT considering the three normal large intestine locations than just the entire colon and rectum. Therefore, it seems important to study the tissue expression of potential biomarkers not only in tumor tissues, but also in NAT and in different sections of large intestine (proximal colon, distal colon, and rectum) for colorectal carcinogenesis.

In summary, our RT-qPCR study showed significantly reduced SST mRNA production in CRC vs. control. A negative correlation between SST mRNA expression and age was observed in both groups. Higher expression of SST2, SST3, and SST5 transcripts was noted in CRC vs. control mucosa. Numerous significant positive correlations were observed between the reciprocal expression of SSTR transcripts in CRC and control. In particular, the increased expression of SST4 mRNA significantly dominated over the other SSTRs transcripts in both study groups (CRC and C). This finding requires further research. The correlation between mRNA and peptide expression was only observed for SST1 in CRC.

The IHC study confirmed the reduced SST expression vs. control and coexpression of all SSTRs in the three tissue groups studied. The expression of SSTRs was heterogeneous, quantitatively diverse, and dependent on cellular localization in the tissue groups studied. Higher expression of SST1, SST2, and SST5 was noted in CRC compared to the control mucosa. Numerous positive correlations were observed between the reciprocal immunoexpression of SSTRs. The dominant SSTRs, both in terms of frequency and intensity of IHC reaction in all the groups studied, were SST1 and SST5. The membranous or membrane-cytoplasmic reaction pattern was dominant for SST2 expression. Immunoexpression of SST2 and SST3 showed the most correlations with clinicopathological data in CRC patients. Overexpression of the SST2 peptide in CRC was correlated with distant hepatic metastasis, while SST3 was associated with a reduced tumor proliferative index. Increased SST3 mRNA expression and decreased SST1-4 immunoexpression were observed in the mucinous vs. nonmucinous subtype of CRC. Our current study did not confirm the prognostic role of the expression of SRIF system components (mRNA and peptide) in either CRC or NAT tissues. This may be due to the homogeneity of the tissue material and requires further studies on larger material.

## 5. Conclusions

Reduced SST expression in CRC indicates a weakening in the antitumor effect of this neuropeptide in this cancer in vivo.Decreased SST production with age may be a risk factor for the development of CRC.Overexpression of SST2 and SST5 in CRC suggests that these receptors play an important role in the pathogenesis of this cancer.Analysis of SST1-5 tissue expression allows for differentiation between the mucinous and nonmucinous CRC subtypes.The coexpression of all SST1-5 and overexpression of not only SST2 and SST5 in CRC may have applications for future therapy based on the SRIF system in sporadic CRC.

## Figures and Tables

**Figure 1 cancers-16-03584-f001:**
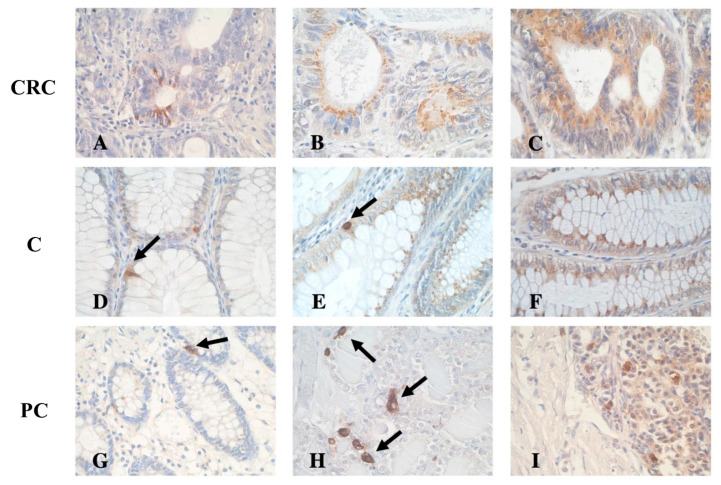
Representative image of the immunohistochemical detection of somatostatin (SST) in different fragments from colorectal cancer (CRC), normal colorectal mucosa adjacent to the tumor (C), and non-colorectal cancer tissues (positive control, PC). (**A**) Focally localized SST-immunoreactive colorectal adenocarcinoma cells with the cytoplasmic IHC reaction; (**B**) a granular IHC reaction in the apical part of the neoplastic cells; (**C**) a strong SST expression in the whole cell cytoplasm of malignant cells; (**D**) SST expression in a few EECs (arrow); (**E**) SST-immunoreactive EECs (arrow) and other positive intestinal epithelial cells in control colon mucosa; (**F**) numerous SST-positive cells in control colonic epithelium; (**G**) SST-immunoreactive EECs (arrow) in colon sample from patient without CRC; (**H**) mature rat thyroid gland with SST-positive C cells (arrows); (**I**) typical carcinoid of the lung with SST expression. New polymer-based immunohistochemistry with DAB staining. Hematoxylin-counterstained. Original magnification ×400.

**Figure 2 cancers-16-03584-f002:**
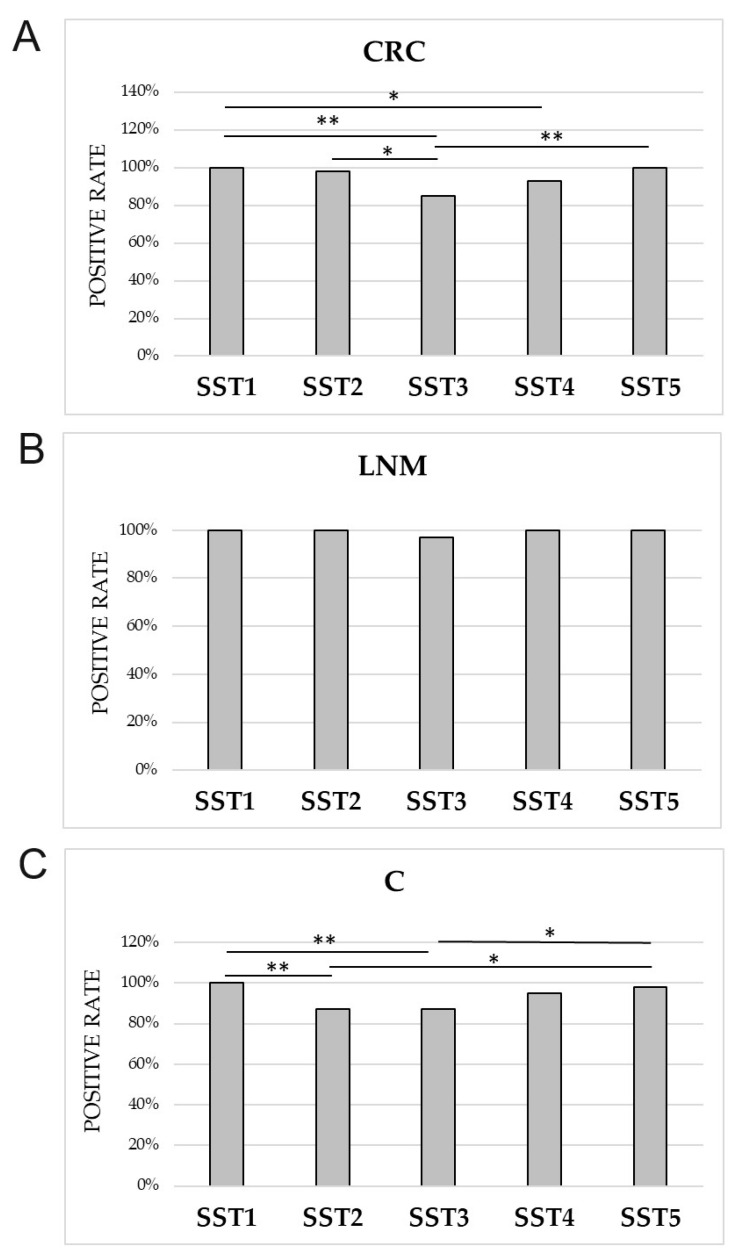
The positive rates of immunoexpression of SST1-5 (%) within different study groups of samples. (**A**) Frequency of SST1-5 expression in colorectal cancer (CRC); (**B**) frequency of SST1-5 expression in lymph node metastasis (LNM); (C) frequency of SST1-5 expression in control colorectal mucosa (**C**). * *p* < 0.05; ** *p* < 0.01.

**Figure 3 cancers-16-03584-f003:**
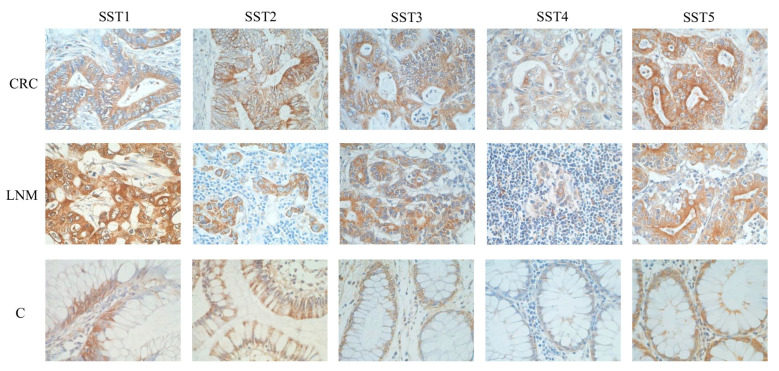
Representative images of the immunohistochemical detection of somatostatin receptors (SST1-5) in colorectal cancer (CRC), lymph node metastasis (LNM), and histologically normal colon crypts (C). Brown staining indicates positive SSTR expression representing different patterns of IHC reaction (cytoplasm/cell membranes/cell nuclei). New polymer-based immunohistochemistry with DAB staining. Hematoxylin counterstained. Original magnification ×400.

**Figure 4 cancers-16-03584-f004:**
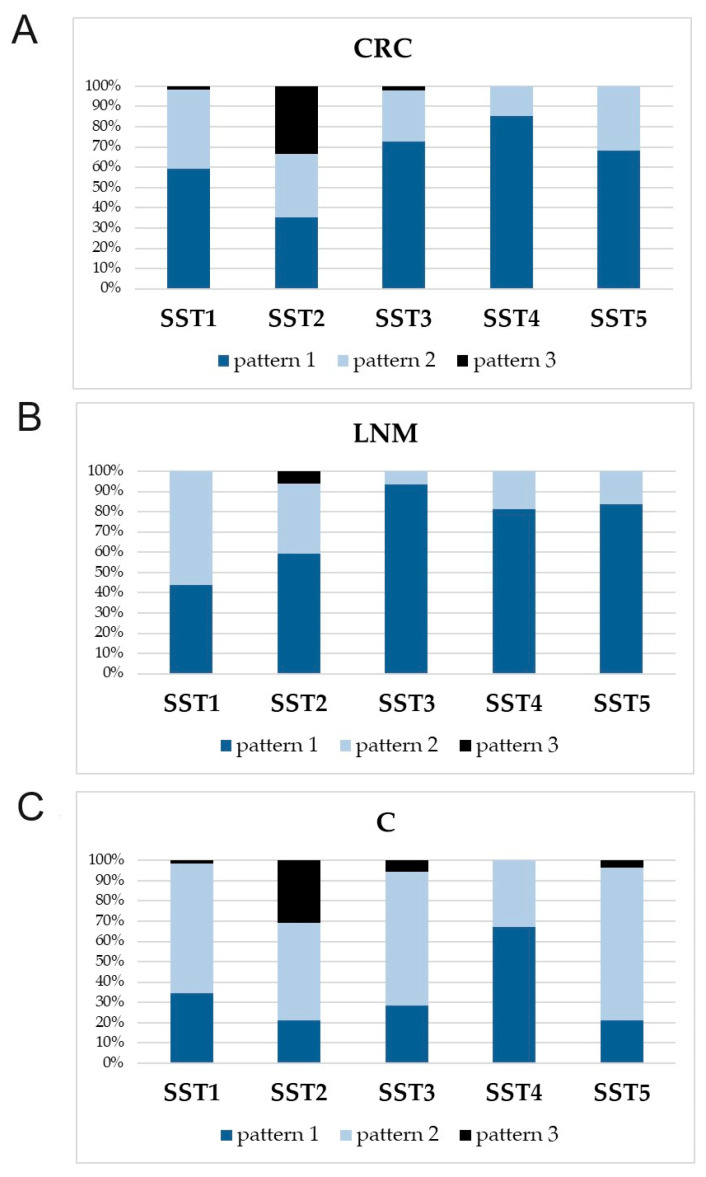
Different patterns of immunohistochemical (IHC) reactions in relation to the cellular localization of SST1-5 within the tested group of samples. (**A**) Percentage distribution of SST1-5 IHC reaction patterns in colorectal cancer (CRC) samples; (**B**) percentage distribution of SST1-5 IHC reaction patterns in lymph node metastasis (LNM); (**C**) percentage distribution of SST1-5 IHC reaction patterns in control mucosa (C). Pattern 1: cytoplasmic and/or cytoplasmic pattern with cell nuclei; pattern 2: mixed pattern with predominance of cytoplasmic over membranous reaction; pattern 3: mixed pattern with predominance of membranous over cytoplasmic reaction and/or clear membranous reaction.

**Figure 5 cancers-16-03584-f005:**
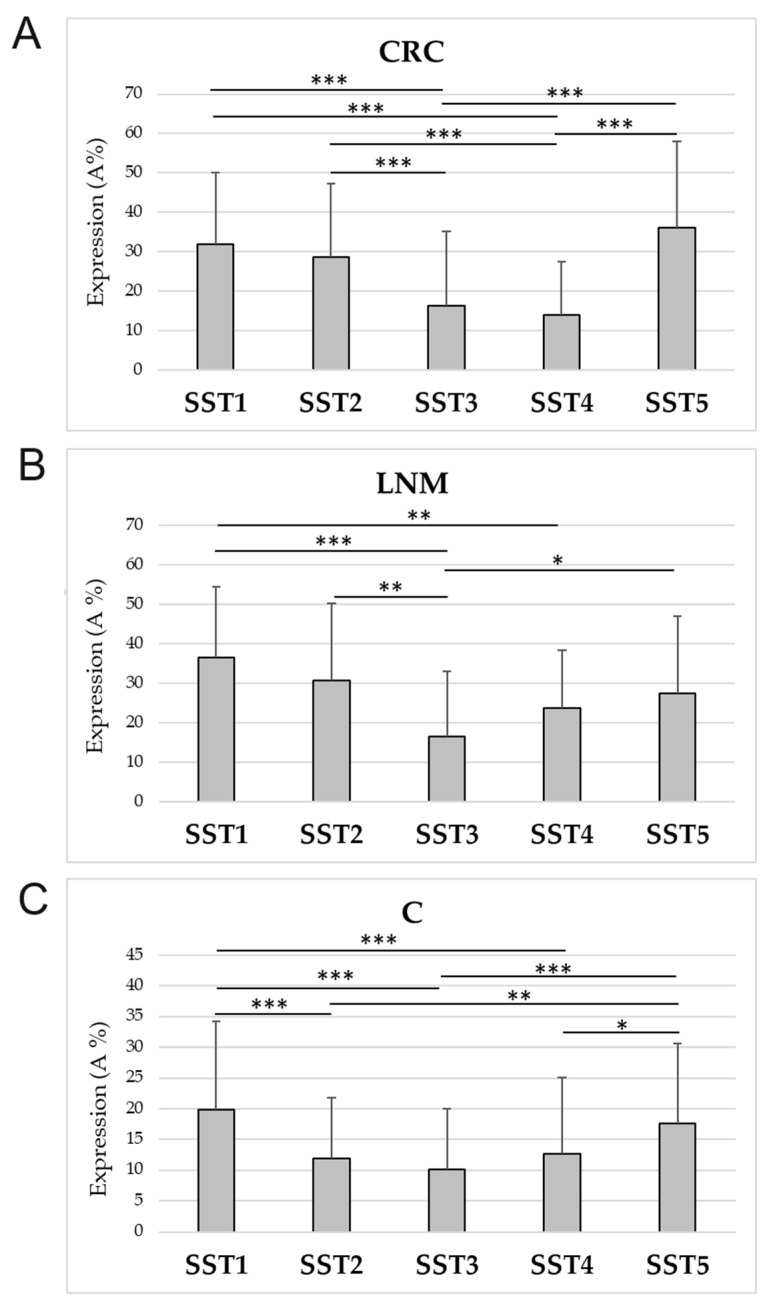
Comparative IHC expression of SST1-5 (mean ± SD) within different study groups of samples. (**A**) SST1-5 expression in colorectal cancer (CRC); (**B**) SST1-5 expression in lymph node metastasis (LNM); (**C**) SST1-5 expression in control mucosa (C). * *p* < 0.05, ** *p* < 0.01, *** *p* < 0.001.

**Figure 6 cancers-16-03584-f006:**
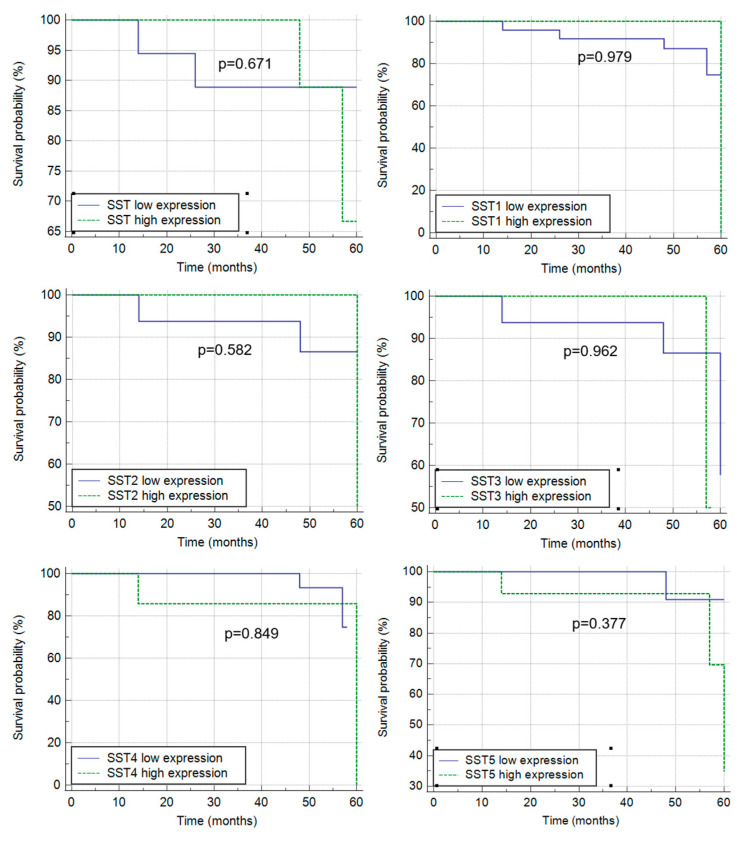
Kaplan–Meier survival curves for CRC patients, in relation to SST and SST1-5 showing that tissue immunoexpression of all components of the SRIF system in tissue samples are not associated with survival time.

**Table 1 cancers-16-03584-t001:** Clinicopathological characteristics of CRC patients at diagnosis.

Age (ys)	Mean ± SD (min–max)—65.44 ± 11.99 (32–89)
Sex (*n*)	Female—6; Male—28
Pathology diagnosis	Colon/rectum adenocarcinoma—22/4 Colon/rectum mucinous adenocarcinoma—7/1
Grade (G) of differentiation	Carcinoma in situ—1 WD (G1)—1 MD (G2)—24 PD (G3)—8
TNM	Tis—1 pT1-T2N0—2 pT3-T4N0—13 pT1-T2N1-N2—1 pT3-T4N1-N2—17 pM0—29 pM1—5
Clinical stage of TNM ^a^	Stage 0 (carcinoma in situ)—1 Stage I—2 Stage II (A, B, C)—12 Stage III (A and B)—13 Stage IV (A and C)—6
Survival time (months)	Mean: 45.4 ± 16.5 months
Deceased	7

Descriptions: ^a^: clinical stage of TNM according to the AJCC/UICC; CRC: colorectal cancer; max: maximum; MD: moderately differentiated tumor (G2); min: minimum; N0: lymph node stage 0; N1+: lymph node stage 1 or higher; PD: poorly differentiated tumor (G3); pM0: distant metastasis stage 0; pM1: distant metastasis stage 1; pT1: tumor stage 1; pT2+: tumor stage 2 or higher; SD: standard deviation; Tis: carcinoma in situ; TNM: tumor–node–metastasis staging system; WD: well-differentiated tumor (G1); ys: years.

**Table 2 cancers-16-03584-t002:** Available clinicopathological data of CRC patients collected in TMA.

Age (ys)	Mean ± SD (min-max)—54.67 ± 11.54 (30–72)
Sex (*n*)	Female—8; Male—25
Pathology diagnosis	Colon/rectum adenocarcinoma—23/6 Colon/rectum mucinous adenocarcinoma—2/2
Grade (G) of differentiation	WD (G1)—6 MD (G2)—12 PD (G3)—15
TNM	pT3N1—19 pT3N2—8 pT4N1—2 pT4N2—4 pM0—33
Clinical stage of TNM ^a^	Stage IIIB—29 Stage IIIC—4

Descriptions: ^a^: clinical stage of TNM according to the AJCC/UICC; CRC: colorectal cancer; max: maximum; MD: moderately differentiated tumor (G2); min: minimum; N1+: lymph node stage 1 or higher; PD: poorly differentiated tumor (G3); pM0: distant metastasis stage 0; pT3+: tumor stage 2 or higher; SD: standard deviation; TNM: tumor–node–metastasis staging system; WD: well-differentiated tumor (G1); ys: years.

**Table 3 cancers-16-03584-t003:** Comparison of the quantitative SST and SST1-5 mRNA expression in CRC patients and in histologically normal colorectal mucosa from the same patients (C).

Type of mRNA	Group	*n*	mRNA Expression [Number of mRNA Copies/µg RNA]							*p* ^a^
			Mean	Median	Min	Max	Q1	Q3	SD	
SST	CRC	25	4.05	0.91	0.00	52.07	0.32	4.34	10.32	<0.0001
	C	25	66.64	19.55	0.18	634.10	4.94	59.59	130.80	
SST1	CRC	25	6501.98	327.62	39.26	81,634.28	150.58	451.84	20,021.25	0.464
	C	25	813.19	295.46	46.29	10,138.86	133.30	474.78	2020.07	
SST2	CRC	25	11.40	10.36	2.98	27.28	8.01	12.67	6.06	<0.0001
	C	25	5.69	4.73	1.57	14.44	3.74	6.89	3.28	
SST3	CRC	25	785.78	423.54	15.58	4743.65	136.55	902.53	1067.88	0.0037
	C	25	257.57	99.88	7.09	3235.87	43.18	206.01	632.90	
SST4	CRC	25	46,261.27	4332.62	377.54	216,246.37	1582.95	75,514.84	69,414.89	0.0594
	C	25	28,431.31	1452.15	55.66	192,020.36	319.34	53,415.22	46,214.63	
SST5	CRC	25	2232.14	1219.25	78.17	14,185.43	706.31	2604.38	3011.51	<0.0001
	C	25	248.00	93.40	3.71	2989.50	55.95	158.91	584.16	

Descriptions: ^a^ Mann–Whitney U test—the data represent the mean, max, min, Q1, Q2, and SD of three independent experiments; CRC: colorectal cancer; max: maximum; min: minimum; *n*: number; SD: standard deviation; SST: somatostatin; SST1-5: SST receptors 1–5; Q1: lower quartile; Q3: upper quartile.

**Table 4 cancers-16-03584-t004:** Values of Spearman’s coefficient for correlation between reciprocal expression of SST and SST1-5 mRNA in colorectal cancer samples (CRC) and non-tumor samples (C).

mRNA Expression	Group	mRNA Expression				
		SST1	SST2	SST3	SST4	SST5
SST	CRC	−0.03	−0.08	0.03	0.01	−0.01
	C	**0.69**	−0.23	−0.28	−0.31	−0.29
SST1	CRC	-	0.20	0.21	0.20	−0.14
	C	-	0.28	0.18	0.10	0.06
SST2	CRC	0.20	-	**0.43**	**0.52**	0.32
	C	0.28	-	**0.42**	**0.52**	0.15
SST3	CRC	0.21	**0.43**	**-**	**0.73**	**0.42**
	C	0.18	**0.42**	**-**	**0.53**	**0.84**
SST4	CRC	0.20	**0.52**	**0.73**	-	0.28
	C	0.10	**0.52**	**0.53**	-	**0.41**
SST5	CRC	−0.14	0.32	**0.42**	0.28	-
	C	0.06	0.15	**0.84**	**0.41**	-

Descriptions: Bold numbers denote Spearman’s rank correlation coefficients R, at *p* < 0.05.

**Table 5 cancers-16-03584-t005:** Incidence of somatostatin (SST) and SST receptor (SST1-5) expression in colorectal cancer (CRC), lymph node metastasis (LNM), and control colorectal mucosa (C).

Group	Positive IHC Expression *n*/All Tested Specimens *n* (%)					
	SST	SST1	SST2	SST3	SST4	SST5
CRC	29/33 (88)	59/59 (100)	57/58 (98) ^a^	51/60 (85)	55/59 (93)	60/60 (100)
LNM	nt	32/32 (100)	32/32 (100) ^a^	30/31 (97)	32/32 (100)	31/31 (100)
C	28/32 (88)	58/58 (100)	52/60 (87)	53/61 (87)	55/58 (95)	57/58 (98)

Descriptions: ^a^ the difference test between two proportions: *p* = 0.024 between CRC and C; *p* = 0.033 between LNM and C; IHC: immunohistochemical; *n*: number; nt: not tested.

**Table 6 cancers-16-03584-t006:** Detection of IHC expression pattern of SST receptors (SST1-5) in colorectal cancer (CRC), lymph node metastasis (LNM), and control colorectal mucosa (C).

Group	IHC Pattern	Expression Pattern *n*/Total Positive Specimen *n* (%)					*p* ^a^
		SST1	SST2	SST3	SST4	SST5	
CRC	1	35/59 (59)	20/57 (35)	37/51 (72)	47/55 (85)	41/60 (68)	<0.001
	2	23/59 (39)	18/57 (32)	13/51 (25)	8/55 (14)	19/60 (32)	
	3	1/59 (0.02)	19/57 (33)	1/51 (0.02)	0	0	
LNM	1	14/32 (44)	19/32 (59)	28/30 (93)	26/32 (81)	26/31 (84)	<0.001
	2	18/32 (56)	11/32 (34)	2/30 (0.06)	6/32 (19)	5/31 (16)	
	3	0	2/32 (0.06)	0	0	0	
C	1	20/58 (34)	11/52 (21)	15/53 (28)	37/55 (67)	12/57 (21)	<0.001
	2	37/58 (64)	25/52 (48)	35/53 (66)	18/55 (33)	43/57 (75)	
	3	1/58 (0.02)	16/52 (31)	3/53 (0.06)	0	2/57 (0.04)	

Descriptions: ^a^ chi-square test (χ^2^), for CRC-χ^2^ = 83.47, for LNM-χ^2^ = 32.65, and for C-χ^2^ = 77.05; 1: cytoplasmic IHC pattern and/or cytoplasmic pattern with cell nuclei; 2: mixed pattern with predominance of cytoplasmic over membranous reaction; 3: mixed pattern with predominance of membranous over cytoplasmic reaction and/or clear membranous reaction; IHC: immunohistochemical; *n*: number.

**Table 7 cancers-16-03584-t007:** Comparison of the quantitative SST and SST1-5 immunoexpression in colorectal cancer (CRC), lymph node metastasis (LNM), and control mucosa (C) (A%).

Type of Peptide	Group	*n*	Area Fraction [%]							*p* ^a^
			Mean	Median	Min	Max	Q1	Q3	SD	
SST	CRC	33	4.04	3.03	0.00	13.19	1.34	5.84	3.72	0.042
	C	32	7.20	7.18	0.00	25.81	1.82	10.46	6.21	
SST1	CRC	59	31.99	30.21	2.03	70.79	16.32	47.50	18.07	<0.0001
	LNM	32	36.57	38.73	5.50	67.64	24.79	46.74	17.93	
	C	58	19.89	16.71	0.18	53.89	8.88	29.79	14.31	
SST2	CRC	58	28.69	28.43	0.00	74.81	12.31	41.13	18.63	<0.0001
	LNM	32	30.63	33.25	3.51	66.54	12.83	43.76	19.60	
	C	60	11.93	9.91	0.00	42.51	3.95	17.21	9.79	
SST3	CRC	61	16.35	9.06	0.00	78.99	1.39	28.42	18.92	0.2855
	LNM	31	16.58	12.34	0.00	69.22	4.54	24.93	16.42	
	C	61	10.13	7.85	0.00	40.14	2.96	14.21	9.26	
SST4	CRC	59	13.97	11.78	0.00	72.44	4.12	21.13	13.51	0.0004
	LNM	32	23.75	25.24	1.33	52.90	10.60	32.70	14.69	
	C	58	12.61	9.74	0.00	54.91	2.79	18.55	12.44	
SST5	CRC	60	36.21	36.48	1.22	84.23	18.36	55.19	21.79	<0.0001
	LNM	31	27.45	20.92	1.02	80.30	13.44	41.33	19.53	
	C	59	17.56	15.15	0.00	47.62	5.10	28.67	13.00	

Descriptions: ^a^ Mann–Whitney U test (for SST), Kruskal–Wallis test, and a multiple comparisons test (for SST1-5); max: maximum; min: minimum; *n*: number; SD: standard deviation; SST: somatostatin; SST1-5: SST receptors 1–5; Q1: lower quartile; Q3: upper quartile.

**Table 8 cancers-16-03584-t008:** Values of Spearman’s coefficient for correlation between reciprocal immunoexpression of SST and SST1-5 in colorectal cancer samples (CRC), lymph node metastasis (LNM), and control colorectal mucosa (C).

Peptide Expression	Group	Peptide Expression				
		SST1	SST2	SST3	SST4	SST5
SST	CRC	−0.22	0.12	0.10	0.04	−0.15
	C	0.35	0.16	0.16	0.30	**0.43**
SST1	CRC	-	**0.49**	0.08	**0.48**	**0.28**
	LNM	-	0.05	−0.11	**0.65**	0.02
	C	-	**0.53**	**0.37**	**0.58**	**0.30**
SST2	CRC	0.49	-	**0.44**	**0.52**	0.12
	LNM	0.05	-	0.31	0.20	**0.78**
	C	**0.53**	-	**0.54**	**0.55**	**0.50**
SST3	CRC	0.08	**0.44**	-	0.20	**0.38**
	LNM	−0.11	0.31	-	0.23	**0.57**
	C	**0.37**	**0.54**	-	**0.45**	**0.51**
SST4	CRC	**0.48**	**0.52**	0.20	-	**0.55**
	LNM	**0.65**	0.20	0.23	-	0.26
	C	**0.58**	**0.55**	**0.45**	-	**0.40**
SST5	CRC	**0.28**	**0.49**	**0.38**	**0.55**	-
	LNM	0.02	**0.78**	**0.57**	0.26	-
	C	**0.30**	**0.50**	**0.51**	**0.40**	-

Descriptions: Bold numbers denote Spearman’s rank correlation coefficients R, at *p* < 0.05.

**Table 9 cancers-16-03584-t009:** Comparison of SST1-5 expression in relation to different IHC patterns of cellular localization.

Group	Pattern	Area Fraction [%]				
		SST1	SST2	SST3	SST4	SST5
CRC		[mean ± SD] [median]	[mean ± SD] [median]	[mean ± SD] [median]	[mean ± SD] [median]	[mean ± SD] [median]
	1	29.2 ± 19.7	35.6 ± 23.0	14.1 ± 14.7	13.6 ± 11.3	34.1 ± 21.6
		26.8	38.4	9.1	10.5	33.1
	2	35.8 ± 15.2	30.8 ± 14.5	32.0 ± 22.1	23.1 ± 21.4	40.7 ± 22.2
		33.9	30.1	33.0	12.9	42.2
	3	#	20.8 ± 13.2	#	#	#
			19.7			
*p* ^a^		0.186	0.056 ^b^	0.008	0.158	0.255
LNM	1	34.6 ± 20.5	24.1 ± 17.9	16.7 ± 16.9	22.6 ± 15.8	28.2 ± 19.9
		33.7	17.8	10.7	23.6	21.1
	2	38.1 ± 16.1	36.9 ± 18.0	22.9 ± 2.0	28.8 ± 7.4	23.8 ± 19.4
		40.2	37.1	22.9	26.9	17.0
*p* ^a^		0.694	0.038	0.506	0.356	0.540
C	1	11.3 ± 8.6	18.4 ± 11.6	7.1 ± 5.4	11.1 ± 10.3	10.3 ± 8.8
		10.9	14.4	5.8	9.2	6.3
	2	24.7 ± 14.9	14.9 ± 7.7	13.8 ± 9.5	17.9 ± 15.2	19.4 ± 13.2
		21.3	15.8	11.6	14.3	18.3
	3	#	8.8 ± 7.9	#	#	#
			7.2			
*p* ^a^		0.0004	0.011 ^b^	0.009	0.097	0.037

Descriptions: ^a^ Mann–Whitney U test, ^b^ Kruskal–Wallis test and a multiple comparisons test (SST2 in C group); #: insufficient data for statistical evaluation; 1: cytoplasmic and/or cytoplasmic pattern with cell nuclei; 2: mixed pattern with predominance of cytoplasmic over membranous reaction; 3: mixed pattern with predominance of membranous over cytoplasmic reaction and/or clear membranous reaction; SD: standard deviation.

**Table 10 cancers-16-03584-t010:** Values of Spearman’s coefficient for correlation between SST and or SST1-5 (mRNA and peptide) in colorectal cancer (CRC) and control mucosa samples (C).

Peptide Expression	Group	mRNA Expression				
		SST1	SST2	SST3	SST4	SST5
SST	CRC	0.30	−0.09	−0.25	−0.30	−0.03
	C	0.02	0.15	−0.05	−0.14	0.16
SST1	CRC	−0.07	**0.52**	0.32	0.28	**0.51**
	C	0.01	0.36	0.18	**0.44**	0.30
SST2	CRC	0.14	−0.18	−0.08	0.13	0.28
	C	0.11	0.10	−0.16	−0.05	−0.16
SST3	CRC	0.08	−0.36	**−0.61**	−0.08	−0.18
	C	−0.13	−0.28	**−0.51**	−0.11	−0.23
SST4	CRC	0.01	0.25	0.28	−0.09	0.35
	C	−0.07	0.09	0.11	0.02	0.43
SST5	CRC	−0.01	0.12	0.04	0.25	0.26
	C	0.02	0.15	−0.05	−0.14	0.16

Descriptions: Bold numbers indicate values of significant R coefficient (*p* < 0.05).

**Table 11 cancers-16-03584-t011:** Comparison of SST1-5 immunoexpression in colorectal cancer (CRC) and control samples (C) in relation to different anatomical sites of the large intestine.

Group	Site	Area Fraction [%]				
		SST1	SST2	SST3	SST4	SST5
CRC		[mean ± SD] [median]	[mean ± SD] [median]	[mean ± SD] [median]	[mean ± SD] [median]	[mean ± SD] [median]
	P	18.8 ± 15.7	17.4 ± 12.2	8.9 ± 19.2	11.9 ± 8.4	36.3 ± 23.7
		14.7	14.9	0.5	11.4	43.5
	D	28.6 ± 16.2	22.4 ± 13.6	13.6 ± 14.8	10.0 ± 10.9	35.1 ± 28.7
		25.4	23.7	10.3	5.9	30.4
	Rectum	30.4 ± 16.2	20.2 ± 15.1	22.6 ± 22.0	11.6 ± 10.6	32.2 ± 18.3
		32.7	13.7	11.5	7.7	25.9
*p* ^a^		0.226	0.780	0.132	0.764	0.954
C	P	12.7 ± 12.4	7.1 ± 7.9	4.9 ± 8.1	11.0 ± 10.7	17.0 ± 13.1
		11.5	6.6	1.4	10.0	15.2
	D	11.7 ± 6.1	3.6 ± 4.8	3.5 ± 3.6	3.1 ± 3.4	9.4 ± 9.0
		12.0	2.4	2.6	2.1	6.1
	Rectum	22.6 ± 16.4	13.6 ± 9.5	14.7 ± 9.0	15.5 ± 13.8	19.5 ± 14.6
		21.1	9.1	12.9	11.8	13.6
*p* ^a^		0.179	0.004	0.001	0.014	0.081

Descriptions: ^a^ Kruskal–Wallis test, a multiple comparisons test (SST2, SST3, and SST4 in C group); D: distal colon; P: proximal colon; SD: standard deviation.

**Table 12 cancers-16-03584-t012:** Comparison of SST and SST1-5 mRNA expression in colorectal cancer (CRC) in relation to mucinous vs. nonmucinous subtypes of CRC.

Subtype of CRC	mRNA Expression [Number of mRNA Copies/µg RNA]					
	SST	SST1	SST2	SST3	SST4	SST5
	[Mean ± SD] [Median]	[Mean ± SD] [Median]	[Mean ± SD] [Median]	[Mean ± SD] [Median]	[Mean ± SD] [Median]	[Mean ± SD] [Median]
Nonmucinous	2.2 ± 2.8	8894.3 ± 23,329.0	11.4 ± 6.3	555.4 ± 761.9	46,887.3 ± 75,783.7	2492.8 ± 3434.5
	0.9	342.1	9.5	257.1	3298.8	1252.4
Mucinous	8.8 ± 19.2	350.2 ± 236.4	11.3 ± 5.9	1378.1 ± 1532.4	44,651.5 ± 54,746.8	1561.8 ± 1458.5
	1.05	314.3	12.3	902.5	5020.6	912.1
***p* ^a^**	0.657	0.495	0.615	0.029	0.458	0.929

Descriptions: ^a^ Mann–Whitney U test; SD: standard deviation.

**Table 13 cancers-16-03584-t013:** Comparison of SST and SST1-5 immunoexpression in colorectal cancer (CRC) in relation to mucinous vs. nonmucinous CRC subtypes.

Subtype of CRC	Area Fraction [%]					
	SST	SST1	SST2	SST3	SST4	SST5
	[Mean ± SD] [Median]	[Mean ± SD] [Median]	[Mean ± SD] [Median]	[Mean ± SD] [Median]	[Mean ± SD] [Median]	[Mean ± SD] [Median]
Nonmucinous	4.8 ± 3.9	35.7 ± 17.8	31.2 ± 19.1	17.7 ± 18.7	16.3 ± 13.9	38.2 ± 21.8
	3.4	40.2	30.9	10.6	12.7	38.0
Mucinous	2.2 ± 2.5	17.5 ± 10.0	19.0 ± 13.1	10.4 ± 19.5	5.0 ± 6.3	27.7 ± 20.8
	0.98	16.6	22.6	0.46	3.0	25.2
***p* ^a^**	0.071	0.002	0.036	0.022	0.0007	0.204

Descriptions: ^a^ Mann–Whitney U test; SD: standard deviation.

**Table 14 cancers-16-03584-t014:** Values of Spearman’s coefficient for correlation between SST and SST1-5 mRNA in colorectal cancer (CRC), control colorectal mucosa (C), age of the patients, and basic laboratory tests.

Type of mRNA	Group	Age (yrs)	Glucose (mg/dL)	WBC (×10^9^/L)	Total Protein (g/dL)
SST	CRC	**−0.42**	0.07	0.09	−0.50
	C	**−0.42**	−0.15	0.31	1.00
SST1	CRC	0.04	0.20	0.09	−1.00
	C	−0.11	−0.24	**0.44**	−0.50
SST2	CRC	0.24	0.01	−0.16	−0.50
	C	0.20	0.02	−0.15	−0.50
SST3	CRC	0.06	0.09	0.07	−0.50
	C	0.03	−0.25	0.26	−1.00
SST4	CRC	0.12	0.06	0.07	−0.50
	C	0.11	−0.13	0.07	−1.00
SST5	CRC	0.08	0.33	0.05	1.00
	C	0.29	−0.33	0.29	−0.50

Descriptions: bold numbers indicate values of significant R coefficient (*p* < 0.05); WBC: white blood cells.

## Data Availability

Data are contained within the article and Appendix A.

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
