# Peer review of "Differentially Expressed Somatostatin (SST) and Its Receptors (SST1-5) in Sporadic Colorectal Cancer and Normal Colorectal Mucosa"

_cancers, 2024, doi:10.3390/cancers16213584_

Round 1
Reviewer 1 Report
Comments and Suggestions for Authors Differentially expressed somatostatin (SST) and its receptors (SST1-5) insporadic colorectal cancer and normal colorectal mucosa, examines levels of somatostatin
and its receptors in a fairly exhaustive analysis across primary and metastatic tumors.
I have a few questions and comments. First, what is meant by non-SST2, lines, 29 and 45?
Lines 119 and 120, a difference is noted in the distance of the control samples for INC and molecular analyses.
I recall from a study on esophageal cancer, that proximal and distal normal were different between each for
expression and histology. Is this your experience in CRC? Please, comment in text.
In section 2.3, what caused the need to toss samples.
Line 148, I assume 60 degrees, 30s is the annealing step, please add the correct extension temp and time.
Line 152, please explain the calculation to get to copy number.
section 2.4, line 160, 5 m thick?
line 171, 40 degrees C.
Lines 178, 179, supplemented? replaced
I am a bit confused over Fig. 7B SST4 and Fig.4.SST4 Fig. 4 does not seem representative
of what I see in the graph.
Comments on the Quality of English Language
Could use some more editing.
Author Response
Comments and Suggestions for Authors
Differentially expressed somatostatin (SST) and its receptors (SST1-5) in
sporadic colorectal cancer and normal colorectal mucosa, examines levels of somatostatin
and its receptors in a fairly exhaustive analysis across primary and metastatic tumors.
Dear Reviewer,
We wish to thank you very much for a review, and time spent on reviewing the manuscript. Thank you and we really appreciate such a favorable review of our work. We tried to address all the comments and errors that arose during the processing of the paper from Word to pdf template, and which we did not correct earlier.
I have a few questions and comments.
First, what is meant by non-SST2, lines, 29 and 45?
Thank you very much for this comment. The terms non-SST2 and non-SST5 referred to receptors other than SST2 and SST5. After some thought, the entire sentence was rewritten.
Lines 119 and 120, a difference is noted in the distance of the control samples for INC and molecular analyses. I recall from a study on esophageal cancer, that proximal and distal normal were different between each for
expression and histology. Is this your experience in CRC? Please, comment in text.
Thank you very much for this comment.
Control tissues for RT-qPCR reactions and immunohistochemistry were evaluated in material collected at a distance of maximum ~10-15 cm and at least 2 cm from the tumor margin, respectively. The morphologically normal mucosa adjacent to the tumor (NAT) in the same patient were also collected in the TMAs. According to the literature, such a distance of control tissues of NAT material (min. 2 cm, max 10 cm) is allowed and required [Ng L, Cells, 2022].
In addition, similar controls (NAT tissues) were available to the authors of the cited papers, hence it was possible to relate our results to those papers. We have completed these discrepancies in the text of the paper.
In the current study, there were no differences in the expression of mRNA of the markers of the SRIF system between the whole colon and rectum. The only significant difference was in SST3 immunoexpression between whole colon and rectum in control tissues. No prognostic role has been demonstrated for SRIF markers in control tissues. All these results are supplemented in the paper of Section 3.5.2 and 3.6, respectively.
In the Results section, we have completed one table (Table 13) on the expression of SSTRs markers depending on the location of the tumor and in the NAT depending on the anatomical section of the colon. In the Introduction and Discussion, we have completed the information on the interesting problem of marker expression in control tissues in different colonic sections and the definition and criteria of NAT itself. Due to the volume of the text, this is only signaled. Thank you for drawing attention to such an important issue also in the CRC.
In section 2.3, what caused the need to toss samples.
Thank you very much for this comment. CRC fragments and control colorectal mucosa from only 25 patients were selected for RT-qPCR analysis because not all 34 patient samples yielded sufficient RNA quality in all primary tumor sites and/or control tissues. The text fragment was completed and explained.
Line 148, I assume 60 degrees, 30s is the annealing step, please add the correct extension temp and time.
Thank you very much for this comment. The text fragment was corrected and completed in the text.
Line 152, please explain the calculation to get to copy number.
Thank you very much for this comment. A description of the calculations obtained by RT-qPCR was completed and provided.
section 2.4, line 160, 5 m thick?
Thank you very much for this comment. The text fragment was corrected and completed in the text. This error occurred when the text was entered into the template.
line 171, 40 degrees C.
Thank you very much for this comment. The text fragment was corrected and completed in the text. This error occurred when the text was entered into the template.
Lines 178, 179, supplemented? Replaced
Thank you very much for this comment. The text fragment was corrected and completed in the text.
I am a bit confused over Fig. 7B SST4 and Fig.4.SST4 Fig. 4 does not seem representative
of what I see in the graph.
Thank you very much for this comment. As per your recommendation, we replaced Fig. 4B with another LMN fragment with SST4 expression. It should be added that the photographs are only with an exemplary but representative in the sense of a proper IHC reaction illustration in a given patient, while to obtain the average in the graph, we first analyzed the average expression from 10 microscopic images, and then the average from all patients and TMA slides, as described in the Materials and Methods section.
The paper has also been read carefully once again, the English language has been corrected by a person who is certified in English and is the first author of the paper (A.G.). We also used the English language correction programs generally available on the Internet (Grammar Checker & Rephraser, among others - https://www.grammarcheck.net/editor/).
All changes (and all additions) in the text were marked red.
Thanking you once again for your efforts to read the work, valid comments, I ask you to be gracious to our efforts and positively accept the changes made.
The best regards,
Aldona Kasprzak

Reviewer 2 Report
Comments and Suggestions for Authors
Somatostatin (SST) prevents the production of hormones from the endocrine system such as growth hormone, thyroid stimulating hormone and prolactin. It binds to a G-protein coupled receptors called SSTRs (SSTR1-5). Anti-tumor effects of SST has been documented in literature. In the present study, the authors have studied the differences in the expression of SST and SSTRs in tissues of control colorectal mucosa, sporadic CRC and in the lymph nodes of metastatic cancer patients. They observed that there is a reduced SST expression in CRC patient tissues, indicating that it may contribute to the weakening anti-tumor effect. At the same time, they also observed that SST2 and SST5 are overexpressed in CRC patients suggesting that these receptors may play a role in the pathogenesis of the CRC.
Comments:
1. The authors state that the observations made in this paper is useful in developing new forms of SST analogs for the treatment of sporadic CRC. This is not explained in this research paper.
Author Response
Somatostatin (SST) prevents the production of hormones from the endocrine system such as growth hormone, thyroid stimulating hormone and prolactin. It binds to a G-protein coupled receptors called SSTRs (SSTR1-5). Anti-tumor effects of SST has been documented in literature. In the present study, the authors have studied the differences in the expression of SST and SSTRs in tissues of control colorectal mucosa, sporadic CRC and in the lymph nodes of metastatic cancer patients. They observed that there is a reduced SST expression in CRC patient tissues, indicating that it may contribute to the weakening anti-tumor effect. At the same time, they also observed that SST2 and SST5 are overexpressed in CRC patients suggesting that these receptors may play a role in the pathogenesis of the CRC.
Dear Reviewer,
We wish to thank you very much for a review, and time spent on reviewing the manuscript. Thank you and we really appreciate such a favorable review of our work.
Comments:
The authors state that the observations made in this paper is useful in developing new forms of SST analogs for the treatment of sporadic CRC. This is not explained in this research paper.
Thank you very much for this comment. The purpose of the study was not to evaluate the effect of specific somatostatin analogs (SSAs) in creating new forms of therapy based on SRIF system components. Our study group did not include patients treated with SSAs, but only with surgical removal of the tumor.
Our concern was that demonstration of both coexpression of all SSTRs or overexpression of many of them (including those other than those described so far) could guide further research in creating other (new) forms of therapy in sporadic CRC based on these peptides. As recommended by the Reviewer, some explanations of the action of SSAs have been completed (in Introduction and Discussion). These explanations are highlighted in blue.
The best regards,
Aldona Kasprzak

Round 2
Reviewer 1 Report
Comments and Suggestions for Authors
The authors addressed my comment adequately.
Author Response
Dear Reviewer,
We wish to thank you very much for review of a revised version of our manuscript, and time spent on reviewing the manuscript.
Thank you for accepting the substantive changes that we made to the manuscript thanks to your comments.
Thank you very much for the re-review sent and for accepting our changes. We really appreciate it.
The best regards,
Aldona Kasprzak and co-authors

Reviewer 2 Report
Comments and Suggestions for Authors
Somatostatin (SST) prevents the production of hormones from the endocrine system such as growth hormone, thyroid stimulating hormone and prolactin. It binds to a G-protein coupled receptors called SSTRs (SSTR1-5). Anti-tumor effects of SST has been documented in literature. In the present study, the authors have studied the differences in the expression of SST and SSTRs in tissues of control colorectal mucosa, sporadic CRC and in the lymph nodes of metastatic cancer patients. They observed that there is a reduced SST expression in CRC patient tissues, indicating that it may contribute to the weakening anti-tumor effect. At the same time, they also observed that SST2 and SST5 are overexpressed in CRC patients suggesting that these receptors may play a role in the pathogenesis of the CRC.
Author Response

(The authors gave the same response as above.)
